# The uptake of population size estimation studies for key populations in guiding HIV responses on the African continent

Nikita Viswasam[1]*, Carrie E. Lyons[1], Jack MacAllister[2], Greg Millett[3], Jennifer Sherwood[3], Amrita Rao[1], Stefan D. Baral[1], on behalf of the Global.HIV Research Group[1¶]

**1** Department of Epidemiology, Key Populations Program, Center for Public Health and Human Rights, Johns Hopkins Bloomberg School of Public Health, Baltimore, MD, United States of America, **2** JWM Consulting, LLC, Washington, DC, United States of America, **3** The Foundation for AIDS Research, amfAR, Washington, DC, United States of America

¶ Membership of the Global. HIV Research Group is provided in the Acknowledgments.
* nviswas1@jhu.edu

**Data Availability Statement:** Only secondary data identified as part of the review are described with their primary sources cited in this manuscript.

## Abstract

### Background

There has been a heightened emphasis on prioritizing data to inform evidence-based HIV responses, including data focused on both defining the content and scale of HIV programs in response to evidence-based need. Consequently, population size estimation (PSE) studies for key populations have become increasingly common to define the necessary scale of specific programs for key populations. This study aims to assess the research utilization of these size estimates in informing HIV policy and program documents across the African continent.

### Methods

This study included two phases; Phase 1 was a review of all PSE for key populations, including men who have sex with men (MSM), female sex workers (FSW), people who use drugs (PWUD), and transgender persons in the 54 countries across Africa published from January 2009—December 2017. Phase 2 was a review of 23 different types of documents released between January 2009 –January 2019, with a focus on the US President's Emergency Plan for AIDS Relief (PEPFAR) and The Global Fund to Fight AIDS, Tuberculosis and Malaria investments, for evidence of stakeholder engagement in PSE studies, as well as key population PSE research utilization to inform HIV programming and international HIV investments.

### Results

Of 118 size estimates identified in 39 studies, less than 15% were utilized in PEPFAR Country Operational Plans or national strategic health plan documents, and less than 2% in Global Fund Concept Notes. Of 39 PSE studies, over 50% engaged stakeholders in study

Thus, all data referenced can be identified using the references included in the manuscript.

**Funding:** This work was supported by Project SOAR (Cooperative Agreement AID-OAA-A-14-00060) (http://www.projsoar.org/), made possible by the generous support of the American people through the United States President's Emergency Plan for AIDS Relief (PEPFAR) and United States Agency for International Development (USAID). The contents of this paper are the sole responsibility of the authors and do not necessarily reflect the views of PEPFAR, USAID, or the United States Government. Funding for this work was also partially provided by the non-profit foundation amfAR. This funding was awarded to Johns Hopkins University, and no personal awards were received. Authors GM and JS are also affiliated with amfAR, and amfAR provided support in the form of salaries for GM and JS, but did not play a role in study design, secondary data collection, analyses, decision to publish, or preparation of the manuscript. Author JM declares a commercial affiliation to JWM Consulting, LLC, which provided support in the form of salary for author JM, but did not have any additional role in the study design, data collection and analysis, decision to publish, or preparation of the manuscript. The specific roles of these authors are articulated in the 'author contributions' section. Additionally, SB and NV's effort was, in part, supported by NIAID Grant R01 AI136664.

**Competing interests:** Author JM declares a commercial affiliation to JWM Consulting, LLC. This does not alter our adherence to PLOS ONE policies on sharing data and materials. All other authors have no competing interests to declare.

implementation and identified target population stakeholders, a third of studies identified policy or program stakeholders, and 15% involved stakeholders in study design.

## Conclusion

The past decade has seen an increase in PSE studies conducted for key populations in more generalized HIV epidemic settings which involve significant investments of finances and human resources. However, there remains limited evidence of sustained uptake of these data to guide the HIV responses. Increasing uptake necessitates effective stakeholder engagement and data-oriented capacity building to optimize research utilization and facilitate data-driven and human rights-affirming HIV responses.

## Introduction

Among both concentrated and generalized epidemics around the world, key populations including female sex workers (FSW), cisgender men who have sex with men (MSM), people who use drugs (PWUD), and transgender women bear a disproportionate burden of HIV when compared to other reproductive-aged adults [1–7]. Specifically, the Joint United Nations Programme on HIV and AIDS (UNAIDS) estimated that between 40% and 95% of new HIV infections in various global regions in 2017 were among key populations and their immediate sexual partners [8]. However, key populations face intersectional stigmas related to key population status and HIV, and thus are often marginalized and undercounted through traditional HIV surveillance [6, 9]. This may be due, in part, to limited coverage of key populations in national surveys, and limited disclosure in surveys of proximal HIV risks, such as sexual practices, that can suggest key population status. Thus, data are often missing for individuals most at risk of HIV acquisition and transmission across settings, potentially limiting data-driven resource allocation and program planning [10]. Key populations also face barriers to engagement in the HIV care continuum leading to lower retention in care and treatment [11–13]. The UNAIDS Gap Report noted this gap in service coverage on a global scale, partly attributed to stigmas in health care settings combined with a lack of key population-specific HIV programs [14].

The Global Fund to Fight AIDS, Tuberculosis and Malaria 2017–2022 Strategy and the US President's Emergency Plan for AIDS Relief (PEPFAR) 3.0 prioritized the generation of data that will better inform an effective HIV response [15, 16]. This has included data focused on both defining the content and strategy of HIV programs, as well as the scale of these programs in response to evidence-based need. The UNAIDS 2016–2021 Strategy has made one of their ten targets to reach 90% of key populations globally with tailored HIV combination prevention and treatment programs[17]. Population size estimates (PSE) for key populations provide denominators in coverage assessments and can facilitate prioritization of public health programs, resource allocation, intervention planning, and evaluation. Moreover, the population denominators provided by PSE can be used in partnership with HIV care continuum and service data to evaluate current coverage of care, treatment retention, and viral suppression, which can identify coverage gaps in order to directly inform the development of evidence-based service targets [13]. Various methods for the empiric measurement of the size of marginalized and key populations have been developed and disseminated through guidelines by UNAIDS and the World Health Organization (WHO) [18, 19].

A growing number of PSE studies have been undertaken for key populations over the past decade [18, 20]. Despite this, there are still many countries where PSE for key populations are inaccurate, incomplete, or missing [20, 21]. A review assessing PSE for key populations among low and middle-income countries revealed that out of 54 countries on the African continent, less than half have published size estimation data for any key population, and of those with existing PSE, about half are considered to have nationally adequate estimates [20]. Inaccurate and incomplete data are particularly common in countries where study participants may be at risk of arrest due to punitive laws criminalizing sex work, drug use, or non-heteronormative sexual practices [9]. Various studies, some in tandem with size estimation exercises, have documented a gap between the burden of HIV in key populations and the coverage of HIV services in these populations in several sub-Saharan African countries [13, 22].

Various frameworks and strategies have been developed for understanding stakeholder engagement in research and its relationship with research utilization [23–25]. These involve a focus on engagement of community and governmental stakeholders throughout study design, implementation, interpretation of findings and dissemination, and utilization through development of recommendations or action plans, policy changes, and national programming changes [25]. To our knowledge, these strategies have not been applied to assess the utilization of studies of PSE among key populations. In response, this study aims to assess stakeholder engagement in PSE studies, and the uptake of PSE in HIV policy and program documents spanning the continent of Africa using adapted measures of research utilization and stakeholder engagement.

## Methods

This study included two phases: Phase 1 consisted of a review of PSE for key populations, including FSW, MSM, PWUD, and transgender persons on the African continent; and Phase 2 entailed a review of country-specific program and policy documents for evidence of research utilization and stakeholder engagement of PSE found in Phase 1, in the context of guiding the HIV response. The original PSE study documents were also reviewed for evidence of stakeholder engagement.

### Phase 1: Systematic review of population size estimates of key populations

The review conducted in Phase 1 assessed the availability of PSE for key populations (FSW, MSM, PWUD, and transgender persons) for the 54 countries on the African continent, as part of a larger global systematic review. The parent systematic review sought to identify all HIV-related data for key populations in peer-reviewed and gray literature. The protocol of this parent systematic review, including search strategies, data items, inclusion/exclusion criteria, screening and selection are described in detail elsewhere [26] and registered in the PROPSERO database (CRD42016047259; 28 September 2016). The current PSE review was conducted following the parent systematic review.

The parent review included studies that met the following eligibility criteria: studies of any design that include data that captures the burden and risk of HIV, prevalence, incidence, prevention indicators, treatment cascade, population size estimates, experienced violence and engagement with healthcare systems data among FSW, MSM, PWUD, transgender persons, and incarcerated populations, even if these groups are not the primary focus of the study; Studies released or presented between January 1, 2006, and January 1, 2019 were included, and data from all countries and settings were included; Study data must be published in a peer-reviewed journal, presented as an abstract at a scientific conference, or available on the web from governmental or non-governmental sources. Data identified through this review were

used to build the Global.HIV data repository, a database hosted by the Research Electronic Data Capture (REDCap) application[27]. After undergoing title/abstract screening and full text review, data from eligible studies were abstracted independently by a team of reviewers, two independent reviewers per article, using standardized data abstraction record forms in REDCap. Each REDCap record contains extracted data of interest and study details from one data source. In each record form, data on each indicator were entered into text fields specific to that indicator, for example, "Size estimate 1 (count)". This record structure enables indicator-specific filtering and selection of results during repository-wide data exports. Differences in data abstraction were resolved using REDCap's data comparison tool by a third, independent reviewer.

Following the parent review, Phase 1 of our review was conducted, identifying PSE through a review of abstracted sources in the Global.HIV database. The Phase 1 PSE review was limited to data found in peer-reviewed or gray literature publications from January 1, 2009 through December 29, 2017 to ensure sufficient time for the data to be available and utilized for policy and program documents released between 2017 and 2019. Through the following steps, the REDCap Data Export tool was used to identify all recordsthat were catalogued as containing PSE data for FSW, MSM, PWUD, or transgender populations from any of the 54 African countries. The export tool presented results of the following selected indicator fields of each record, with a filter limiting records to those with data in eligible countries: population of interest, size estimates (absolute count), PSE method, location(s) of PSE, country of data collection, and year of publication. The data source of each record presenting eligible PSE data in these export results then underwent full text review to further extract the operational definition used for each estimated population, the funder or implementer of the PSE study, and the year of estimation. The quality of the PSE identified was not assessed for this study.

## Phase 2: Assessment of research utilization and stakeholder engagement of pse of key populations available in the literature

The results from Phase 1 informed Phase 2 analyses, which sought to determine if PSE identified in Phase 1 informed policy, programming, or resource allocation through evidence of research utilization and stakeholder engagement. Phase 2 represented a review of 23 different types of documents related to HIV policy, programming, and program scale up in African countries, with a focus on PEPFAR and Global Fund investments. The scope of Phase 2 was limited to documents released between January 2009 and January 2019 referencing country settings where PSE data was identified in Phase 1. The types of documents reviewed include PEPFAR Country Operational Plans (COPs); Demographic and Health Surveys (DHS); Global Fund Frameworks; Global Fund Concept Notes; Global Fund Transitional Funding Mechanism (TFM) Proposals; Global Fund Procurement and Supply Management (PSM) plans; Global Fund regional expressions of interest; Global Fund Funding Models; Global Fund Secretariat briefing notes; peer-reviewed documents; PEPFAR country frameworks; PEPFAR reports; MEASURE Evaluation publications; non-governmental organization (NGO) reports; National AIDS/HIV Office documents; National Strategy or Ministry of Health (MOH) documents; UNAIDS progress reports from 2015–2018; UNAIDS epidemiological data publications; UNAIDS surveillance guidelines; United Populations Fund (UNFPA) manuals; and WHO HIV/AIDS guidelines. These documents were identified through searches in the following grey literature databases and organizational websites: POPLINE, USAID Development Experience Clearinghouse, The Global Fund country portfolio site pages, PEPFAR Country Operational Plan online archive, WHO African Region Library, WHO Global Publications Repository, MEASURE Evaluation publications archive, UNAIDS Progress Reports archive,

UNAIDS country publication archives, country-specific Ministry of Health website publication archives, and country-specific National Bureau of Statistics archives.

Documents identified through this search were screened for reference to PSE data. The full text of relevant documents were then reviewed by a single reviewer to determine if any of the PSE found in the Phase 1 review were referenced in the context of research utilization and stakeholder engagement, each identified through a set of indicators. The development of these indicators were informed by research utilization and stakeholder engagement implementation guidelines by Population Council [25]. The indicators, corresponding types of sources assessed, and examples of language reflecting each form of utilization are outlined in Table 1.

## Results

The Phase 1 review produced 118 PSE from 39 studies for key populations in 22 countries in Africa. Overall, 70 PSE were available for FSW, 27 for MSM, 21 for PWUD, and none for transgender persons. All estimates of PWUD identified were exclusively of people who inject drugs (PWID). National, regional, or local PSE were identified for Angola, Burkina Faso, Burundi, Cameroon, Cote d'Ivoire, Egypt, eSwatini, Ethiopia, Ghana, the Gambia, Kenya, Mauritius, Mozambique, Morocco, Niger, Nigeria, Rwanda, Senegal, South Africa, Tanzania, and Togo. In total, 30% (36/118) were national estimates, 52.5% (62/118) were district or city level estimates, and 14.4% (17/118) were provincial estimates. Three regional estimates were available, 2 of which were for Eastern Africa. Overall, 64% (25/39) of studies reporting PSE data were present in the peer-reviewed literature only, and 36% (14/39) were identified in gray literature, five of which had also published results in peer-reviewed articles. Of grey literature reports, 11 included PSE activities as a component of HIV surveillance studies commissioned in collaboration with governmental organizations, two were PSE-specific studies led by governmental organizations, and one was an NGO-led report with mention of PSE data. The outcomes of each stakeholder engagement and research utilization indicator assessed per study are shown in Tables 2 and 3.

### Research utilization

Of the 118 PSE identified, 11% (13/118) of PSE were referenced to change a PEPFAR country program or policy as documented in the COPs of 5 countries including Angola, Cameroon, Ghana, Kenya, and South Africa. Of referenced estimates, 46% (6/13) were national, 46% were city-level estimates, and 8% (1/13) were provincial. In Cameroon, MSM size estimates were used to change geographic prioritization of service coverage and expand service to new areas of the country [51]. In Ghana, FSW size estimates were used to develop area-specific targets for antiretroviral therapy (ART) coverage, set and track progress on FSW programming targets, and inform provision of technical assistance coverage [52]. PSE were referenced in South Africa, Kenya, and Angola COPS to improve service provision for PWID [53, 54], MSM [55] and FSW [54]. Size estimates also informed the decision to plan provision of training and technical assistance in the HIV care continuum in select high-impact sites in Angola [55] and updated prevention guidelines for PWID in South Africa [53].

Of 118 PSE, 1.7% (n = 2) estimates were referenced to change Global Fund program or policy as documented in the Global Fund Concept Notes for Senegal and Kenya from 2014 to 2015. In Kenya, PWID estimates were used to estimate current service coverage and establish targets for program scale up [56], and PWID estimates in Senegal were used to develop outreach and syringe program activities [57]. No PSE for FSW or MSM were referenced in Global Fund documents.

**Table 1. Description of research utilization indicators and stakeholder engagement indicators.**

| Research Utilization Indicators | Types of Documents Reviewed | Evidence Examples |
|---|---|---|
| Have stakeholders developed an Interpretation and Use Plan for size estimation data? | Country Operational Plans (COP), Global Fund (GF) Country Coordinating Mechanism (CCM) concept notes, WHO HIV/AIDS guidelines, Global Fund Procurement and Supply Management (PSM) plans, Global Fund regional expressions of interest, UNAIDS progress reports, Phase 1 PSE study documents | Language around guidance and suggestions for interpretation of data in local context and uses for study for guiding future key population data collection, program development, or service delivery |
| Have size estimation data been used to identify a problem? | COPs, GF CCM concept notes, UNAIDS epidemiological data publications, UNAIDS surveillance guidelines, WHO HIV/AIDS guidelines, National AIDS/HIV Program or Council documents, DHS, peer-reviewed documents, PEPFAR country frameworks, Global Fund PSM plans, UNFPA manuals, UNAIDS 2015 progress reports | Language around gaps in key population HIV response; Scale of services or resources; Contextualizing service utilization data |
| Have size estimation data been used to develop a plan of action/recommendation to address that problem? | COPs, GF Framework, GF CCM concept notes, UNAIDS data publications, National AIDS/HIV Program or Council documents, PEPFAR country frameworks, GF PSM plans, MEASURE Evaluation publications, United Nations Population Fund (UNFPA) manuals, UNAIDS 2015 progress reports, Phase 1 PSE study documents | Using PSE to propose specific changes in level and areas of program coverage or resource allocation |
| Have size estimation data been used to direct service delivery? | COPs, National Strategic Plan documents, GF CCM concept notes, WHO HIV/AIDS guidelines, PEPFAR country frameworks, PEPFAR COPs GF PSM plans, GF regional expressions of interest, MEASURE Evaluation publications, UNAIDS 2015 progress reports | Language linking PSE to proposed direction of resources or services towards or away from key populations |
| Have size estimation data been used to change a Global Fund program or policy as documented in concept notes? | GF concept notes, GF Funding Model, GF Secretariat briefing notes | Language linking size estimates to adjustments in key populations priority areas, program development, program coverage targets, or changes in resource allocation around key population-tailored services |
| Have size estimation data been used to change a PEPFAR program or policy as documented in country operational plans? | PEPFAR COPs, PEPFAR Strategic Direction Summaries | Language linking size estimates to adjustments in key populations priority areas, program development, program coverage targets, or changes in resource allocation around key population-tailored services |
| Have size estimation data been used to change a national MOH program or policy as documented in NSPs? | National Strategic Plan documents (HIV/AIDS strategies, health strategies), GF CCM concept notes, National AIDS/HIV office documents, GF PSM plans, UNAIDS progress reports | Language linking size estimates to adjustments in key populations priority areas, program development, program coverage targets, or changes in resource allocation around key population-tailored services |
| Results/data translated into non-academic resources (briefs/ advocacy tools) | Phase I PSE study documents; Gray literature | Results dissemination briefs, presentations, or advocacy reports |
| **Stakeholder Engagement Indicators** | **Types of Documents Reviewed** | **Evidence Examples** |
| Have stakeholders been identified, who would be needed to make policy/program decisions? | Phase 1 PSE study document, COPs, GF CCM concept notes, Global Fund PSM plans, MEASURE Evaluation publications, UNAIDS epidemiological data publications, UNAIDS progress reports | Language around involvement of National AIDS Council or Ministry of Health members, local program implementing non-governmental organizations (NGOs) and key population community and advocacy groups |
| Have stakeholders been identified that represent the target population? | COPs, GF CCM concept notes, UNAIDS epidemiological data publications, Global Fund PSM plans, MEASURE Evaluation publications, UNAIDS surveillance guidelines, UNAIDS progress reports | Language of key population members involved in study design, implementation, dissemination |
| Have stakeholders been engaged throughout study design? | Phase 1 PSE study document | Language that evidences 1) working with stakeholders (including program implementers, policy makers, and the target population) to identify opportunities for use of the study's data, findings, and recommendations; 2) enhancement of the protocol and study design with stakeholders' knowledge of local context; 3) documentation of the stakeholders' role and responsibilities in the study's research utilization process; and 4) development of stakeholder's capacity to understand and manage research utilization processes |

(*Continued*)

**Table 1.** (Continued)

| Have stakeholders been engaged throughout study implementation? | Phase 1 PSE study document | Involvement of site mapping and participation in estimation exercises (such as peer educators distributing unique objects) |
| --- | --- | --- |
| Were study data/results shared by stakeholders with other groups? | Peer-reviewed academic journals, UNAIDS surveillance guidelines | Language referencing dissemination of study data with other organizations or groups |
| Were stakeholders included as authors on published document? | Peer-reviewed academic journals, Phase 1 PSE study document | - |
| Was a study-specific advisory panel established? | Phase 1 PSE study document | Language referencing a technical working group, advisory panel, community advisory board, or other group of stakeholders established for the purpose of advising study activities |

Overall 6% (7/118) of PSE were referenced to change a national Ministry of Health policy or program, as documented in Strategic Plans among 4 countries, including Ghana, Kenya, Senegal, and South Africa. In Senegal, PWID size estimation led to the integration of PWID as a vulnerable population in its National Strategic Plan in Response to AIDS and investment in an integrated prevention and methadone program in Dakar [57, 58]. In South Africa, PSE for FSW were incorporated the South African National Sex Worker HIV Plan 2016–2019 to define provincial-level programmatic service targets for sex workers [59]. The National Strategic Plan in Kenya referenced PSE for both FSW and PWID to estimate current service coverage and establish targets for program scale up, also informing the development of the National Guidelines for HIV/STI Programming with Key Populations report [60], and the National Strategic Plan for Ghana referenced PSE for FSW, which were used to define a minimum HIV service package for FSW [61].

In total, 41% (48/118) of PSE were referenced to identify a problem. The most common of such problems were gaps in response and/or service coverage, and to highlight the lack of comprehensive data on the target population. FSW were the most frequently mentioned population (38%), with PWID being the least mentioned (27%). Of the PSE used to identify a problem, 29% (14/48) were used to develop a plan of action or develop a recommendation to address the problem for 9 countries, including Angola, Cameroon, Egypt, Kenya, Morocco, Niger, Nigeria, Senegal, and South Africa. Five of the countries referenced one key population in the plans of actions and recommendations, with Niger [62] and Nigeria [63] referencing PSE for FSW; Senegal referencing PSE for PWID [57]; Ghana referencing a national PSE estimate for FSW [61]; and Angola referencing PSE for MSM [55]. Three of the countries referenced PSE for two key populations including: South Africa referencing FSW and PWID [59]; Morocco referencing PSE for MSM and PWID [64]; and Kenya referencing PSE for FSW and PWID [60]. One country, Egypt, referenced PSE for 3 key populations: FSW, MSM, and PWID [65]. In all references used to develop a plan of action, resources proposed were directed towards rather than away from key populations.

Of the 59 PSE identified from the peer reviewed literature, 29% (n = 17) were translated into non-academic resources, including briefs or advocacy tools. The 17 PSE non-academic resources included PSE among 1 region (Eastern Africa) and 8 countries, including Angola, Ethiopia, Ghana, Kenya, Mauritius, Senegal, South Africa, and Tanzania. Kenya was the only country that translated the PSE into non-academic resources for the three key populations for which PSE were identified (FSW, MSM, PWID). Ethiopia, Ghana, Mauritius, and South Africa translated PSE for FSW into non-academic resources. PSE for PWID were translated into non-academic resources in Senegal and Tanzania; and PSE for MSM were translated into non-academic recourses for Angola.

**Table 2. Outcomes of research utilization of and stakeholder engagement in identified size estimate studies.**

| Country | Year of Estimation | Population | Research Utilization | | | | | | | Stakeholder Engagement | | | | | | | |
|---|---|---|---|---|---|---|---|---|---|---|---|---|---|---|---|---|---|
| | | | Stakeholders developed an Interpretation and Use Plan for PSE | PSE used to identify a problem | PSE used to develop a plan of action/recommendation to address that problem | PSE used to change a Global Fund program/policy | PSE used to change a PEPFAR program/policy | PSE used to change a national MOH program/policy | Results/data translated into non-academic resources | Stakeholders identified who would be needed to make policy/program decisions | Stakeholders identified that represent the target population | Stakeholders engaged throughout study design | Stakeholders engaged throughout study implementation | Study objectives align with stakeholder priorities/in-country data needs | Study data/results shared by stakeholders with other groups | Stakeholders included as authors on published document | Study-specific advisory panel established |
| Angola [28] | 2011 | MSM | | | | | | | | | | | | | | | |
| Burundi [29] | 2013 | FSW | | | | | | | | | | | | | | | |
| Cameroon [30] | 2013 | FSW | | | | | | | | | | | | | | | |
| | 2013 | MSM | | | | | | | | | | | | | | | |
| Cote d'Ivoire [31] | 2008 | FSW | | | | | | | | | | | | | | | |
| Egypt [32] | 2014 | FSW | | | | | | | | | | | | | | | |
| | 2014 | MSM | | | | | | | | | | | | | | | |
| | 2014 | PWID | | | | | | | | | | | | | | | |
| Ethiopia [33, 34] | 2010 | FSW | | | | | | | | | | | | | | | |
| | 2013 | FSW | | | | | | | | | | | | | | | |
| eSwatini [35] | 2014 | FSW | | | | | | | | | | | | | | | |
| Gambia, The [36] | 2013 | FSW | | | | | | | | | | | | | | | |
| | 2013 | MSM | | | | | | | | | | | | | | | |
| Ghana | 2011 | FSW | | | | | | | | | | | | | | | |
| Kenya [31, 37–39] | 2008 | FSW | | | | | | | | | | | | | | | |
| | 2010 | FSW | | | | | | | | | | | | | | | |
| | 2010 | MSM | | | | | | | | | | | | | | | |
| | 2010 | PWID | | | | | | | | | | | | | | | |
| | 2012 | FSW | | | | | | | | | | | | | | | |
| | 2012 | MSM | | | | | | | | | | | | | | | |
| | 2012 | PWID | | | | | | | | | | | | | | | |
| | 2011 | FSW | | | | | | | | | | | | | | | |
| | 2011 | FSW | | | | | | | | | | | | | | | |
| | 2011 | MSM | | | | | | | | | | | | | | | |
| | 2011 | PWID | | | | | | | | | | | | | | | |
| Mauritius [40, 41] | 2009 | FSW | | | | | | | | | | | | | | | |
| | 2010 | FSW | | | | | | | | | | | | | | | |

**Table 3. Outcomes of research utilization of and stakeholder engagement in identified size estimate studies, continued.**

| Country | Year of Estimation | Population | Research Utilization | | | | | | | | Stakeholder Engagement | | | | | | | |
|---|---|---|---|---|---|---|---|---|---|---|---|---|---|---|---|---|---|---|
| | | | Stakeholders developed an Interpretation and Use Plan for PSE | PSE used to identify a problem | PSE used to develop a plan of action/recommendation to address that problem | PSE used to change a Global Fund policy as documented in concept notes | PSE used to change a PEPFAR policy as documented in country operational plans | PSE used to change a national MOH policy as documented in NSPs | Study results published in a peer-reviewed academic journal | Results/data translated into non-academic resources (briefs/pamphlets/advocacy tools) | Stakeholders identified who would be needed to make policy/program decisions | Stakeholders identified that represent the target population | Stakeholders engaged throughout study design | Stakeholders engaged throughout study implementation | Study objectives align with stakeholder priorities/in-country data needs | Study data/results shared by stakeholders with other groups | Stakeholders included as authors on published document | Study-specific advisory panel established |
| Morocco [42] | 2010 | FSW | | ▩ | | | | | ▩ | | | | | | | | | |
| | 2010 | MSM | ▩ | ▩ | ▩ | | | | ▩ | | | | ▩ | | | | ▩ | |
| | 2010 | PWID | ▩ | | | | | | ▩ | | | | | | | | ▩ | |
| | 2013 | FSW | | | | | | | ▩ | | | | | | ▩ | | ▩ | |
| | 2013 | MSM | | | | | | | ▩ | | | | | | | | ▩ | |
| | 2013 | PWID | | | | | | | ▩ | | | | | | | | | |
| Mozambique [43] | 2009 | FSW | | | | | | | | | | ▩ | | | | | | |
| Nigeria [44, 45] | 2009 | PWID | | | | | | | | | | | | | | | | |
| | 2009 | MSM† | | | | | ▩ | | | | | | | | | | | |
| | 2012 | FSW | ▩ | ▩ | ▩ | | | | ▩ | | | ▩ | ▩ | ▩ | ▩ | ▩ | ▩ | ▩ |
| Niger [46] | 2011 | FSW | | | ▩ | | | | ▩ | ▩ | | | | | | | | |
| Rwanda [47] | 2010 | FSW | | ▩ | | | | | ▩ | | | | | | | | | |
| Senegal [48] | 2011 | PWID | | | ▩ | ▩ | | ▩ | | | ▩ | | | ▩ | | | | ▩ |
| South Africa [49] | 2012 | FSW | | ▩ | | | | | | | | | | ▩ | ▩ | | ▩ | |
| | 2013 | FSW | | | | | | | | ▩ | | | | | | | | |
| Tanzania [39] | * | PWID | | | | | | | | | | | | | | | | |
| Tanzania (Zanzibar) [50] | 2012 | FSW | | | | | | | | | | | | ▩ | ▩ | | | |
| | 2012 | MSM | | | | | | | | | | | | | | | | |
| | 2012 | PWID | | | | | | | | | | | | | | | | |
| Togo [36] | 2013 | FSW | | | | | | | | | | | | | | | ▩ | |
| | 2013 | MSM | | | | | | | | | | | | | | | | |

*Year of estimation not available in reviewed document

† MSM who engaged in sex work only were included in size estimation activities

## Stakeholder engagement

Overall, 33% (13/39) of PSE publications documented identification of stakeholders who were policymakers or program implementers in Cameroon, Côte d'Ivoire, eSwatini, the Gambia, Ghana, Kenya, Mauritius, Rwanda, Senegal, South Africa, Tanzania, and Togo. In total, 87% (34/39) of PSE studies documented identification of stakeholders that represent the target population. In total, 46% (19/39) of studies reported objectives in alignment with stakeholder or in-country data needs, and 15% (6/39) of studies document an advisory panel established for the study. Overall, 33% (13/19) of studies included stakeholders as authors in the published document. The most common form of stakeholder engagement in assessed studies involved study implementation through the use of key informants for mapping of KP hotspots (n = 6), of peer outreach workers who implemented estimation exercised (n = 5), and of stakeholders convened to interpret findings and develop consensus on estimates (n = 3).

In total, 61.5% (24/39) studies reported stakeholder involvement in implementation while 15% (6/39) studies reported stakeholder involvement in study design, in Morocco, Cameroon, South Africa, Egypt, Côte d'Ivoire, Kenya, and Togo. In Morocco, a workshop was held in Rabat with national stakeholders to interpret and discuss the PSE results [64]. Local key population members were interviewed in Egypt to inform the selection of size estimation methods used in the study based on effectiveness, acceptability, and adaptability to local context [32]. In a study conducted in Côte d'Ivoire and Kenya, mapping used for size estimation was also developed into a tool for local implementing NGO to improve their prevention outreach [31]. In a PSE study in Cameroon, study protocol was submitted to the Ministry of Health for review, and feedback was incorporated into the final protocol [66].

## Discussion

PSE studies for key populations are becoming increasingly common, and these studies have involved significant investment of finances and human resources to undertake. However, the results presented here suggest only a small proportion of such studies are being used to inform programming by country governments and their funding and implementation partners, including PEPFAR and the Global Fund. Additionally, we observed that about a third of PSE studies that were cited in program and policy documents were not utilized by PEPFAR or the Global Fund through any method assessed in this review, such as to identify a problem, develop a plan of action, or inform planning of key populations programming. This speaks to the need to improve the effective accessibility and uptake of size estimation data, which requires not only increased reporting of estimates in planning documents by program implementers, funders and policy makers, but the utilization of estimates to guide the HIV response at the national and regional levels.

PSE of key populations can be used to inform appropriate resource allocation for key populations, programmatic targets, and approaches to HIV prevention and treatment programs with appropriate coverage of key populations. This review found that country PEPFAR documents that utilized estimates did so most commonly as evidence for geographic reprioritization of key population service coverage, targeted technical assistance, service expansion for HIV care, and target setting [51, 52, 55]. In national strategic plans, estimates were utilized to define a minimum HIV service packages for key populations, scale up service coverage, and develop and implement evidence-based interventions [58, 61]. Where estimates have been utilized, the actions that have been taken in response as illustrated above are encouraging [57]. However, there remain substantial gaps in the utilization of research to inform program and policies. In particular, PSE are largely unused in policy and programs documents by PEPFAR and even less so for the Global Fund.

In a review of available size estimates for key populations, Sabin and colleagues noted that while many estimates are developed and submitted by national HIV/AIDS programs to UNAIDS, few estimates published in peer-reviewed literature were endorsed by national authoritative stakeholders, and thus were not used in national planning [20]. The low utilization of estimates in this review, particularly of estimates found in the peer-reviewed literature, reinforces this finding. Close engagement of government stakeholders throughout the process of study design, implementation, and data use may support increased uptake and utilization of estimates in local and national programming. Researchers can also build the capacity of stakeholders to interpret and use study findings in guiding service development and targets; these forms of stakeholder engagement were found to be limited when evaluating size estimation studies. While over half of PSE studies reported some stakeholder involvement in implementation, few studies reported stakeholder engagement in study design. A study-specific advisory panel was established in some studies, as well as stakeholders convened to interpret findings and develop consensus on estimates in others, but there was limited documentation of established strategies to improve research utilization by working with stakeholders on ways to use the studies' findings in stakeholders' own work or develop their capacity for research utilization. The primary strategies of stakeholder engagement seen in this review allow for context-specific interpretation of data after study implementation, and increase accessibility of findings. However, without stakeholder engagement in study design, study teams may be unaware of contextual factors that can influence the quality of study implementation and results, and may limit confidence and endorsement in resulting estimates by stakeholders who make program and policy decisions [31, 32, 64, 66]. The cases reported above present examples of engagement researchers can undertake to involve stakeholder in studies, but ultimately indicate that comprehensive stakeholder engagement was limited to a few studies.

In a systematic review of stakeholder engagement methods reported in effectiveness and outcomes research, Concannon et al noted that stakeholder engagement is most common in early stages of research and least common in the process of dissemination and application of findings [23], similar to the observations of PSE studies assessed in this review. While studies often involved stakeholders in implementation, literature on stakeholder engagement notes that a key component is stakeholder provision of guidance on research dissemination and use of data for future program design and implementation [24]. This is also described as stakeholder engagement impact through immediate, intermediate, and long-term outcomes [24]. Immediate outcomes include informing research questions, methods, interpretation and dissemination; intermediate outcomes include the value and uptake of research; and long-term outcomes involve decision-making and health policy. While frameworks of stakeholder engagement focus on engagement of community stakeholders, in the context of key population PSE, the engagement of stakeholders needed to make policy and program decisions, such as Ministries of Health, National HIV/AIDS programs, as well as local implementing NGOs, play a central role in enacting intermediate and long term outcomes of PSE research. Existing conceptual models of engagement suggest that engaging stakeholders who make program and policy decisions when conducting PSE studies may be tied to utilization of research in program planning. Various resources have been developed to guide researchers and implementers on incorporating stakeholder engagement in research and its utilization [24, 25].

This review is limited by the reporting of information in studies and planning documents. Thus, if indicators of stakeholder engagement were not explicitly reported in a study document, it was not possible to know whether or not these conversations took place. Likewise, if PSE sources were not clearly cited in documents being assessed for research utilization, we concluded that those size estimates were not considered in the policy and programmatic decision-making processes documented. We acknowledge that publication requirements and

report length limits can lead to underreporting of existing stakeholder engagement by studies. Organizations also collect programmatic data that may often go unpublished along with dissemination and advocacy tools, which could not be identified in this review. Furthermore, some data are not publicly available during the development of decision-making documents. In many cases, size estimation is conducted as one component of larger studies on HIV prevalence, risk behaviors, and the care continuum, and recommendations specific to size estimation results may not be reported. As the focus of the second phase of the review was on PEPFAR and Global Fund investments, which emphasized documentation in online archives, this review may have missed unpublished planning documents by other country-level agencies demonstrating PSE research utilization. The Global.HIV systematic review may not have captured all available size estimation sources, as it was limited to sources found online through peer-reviewed literature databases and gray literature. In some PSE studies, underestimations that limit the utility of PSE in informing programming may also play a role in the lack of reference to PSE studies in planning documents. This review did not assess PSE quality, but other reviews have documented that of African countries with existing PSE, about half are considered to have nationally adequate estimates [20].

Overall, most published PSE studies of key populations identified have limited documented utilization in HIV policy and planning documents. This review has highlighted the need to improve effective uptake of size estimation data to guide the HIV response, as well as the need for the development of more size estimates of key populations in both generalized and concentrated epidemic settings on the African continent. Moreover, the near complete absence of size estimates for transgender persons is especially concerning in the context of the high burden of HIV among transgender women [7]. The low proportion of nationally representative estimates found in this review and elsewhere [20] also reinforces the need to improve estimation approaches that yield national-level data, such as small area estimation and extrapolation [67]. Considering the importance of PSE in guiding data-driven HIV responses, the data presented here display an opportunity to build capacity to ensure that available data appropriately guides responses, and that optimal decisions are made about data needs moving forward. Overall, those focused on data-driven responses can establish relationships with country governmental and non-governmental stakeholders, conduct collaborative country-driven data collection, and build their capacity to understand and interpret size estimates and their quality, which can encourage endorsement of estimates and their subsequent utilization in planning. In light of the disproportionate burden of HIV documented among key populations coupled with limited coverage of existing HIV services, the development and utilization of accurate size estimates is ever more crucial to tailor effective and efficient responses to the HIV pandemic.

## Acknowledgments

We acknowledge Erin Sullivan, Shaheen Kurani, and the Global.HIV Research Group, without whom this review would not be possible. The Global.HIV Research Group is made up of the following members: Zafir Abutalib, Chase Alston, Joe Amoah, Anna Bickers, Ashley Charest, Meghan Holtzman, Cameron Meade, Lookman Mojeed, Madeline Nelson, Albert Osei, Mariela Pinedo, Summer Rak, Arlene Reich, Yvonne Robles, Kavya Sanghavi, Alex Schmall, Owen Stokes-Cawley, Vinithra Varadarajan, Kia Vaughn, Dexter Waters.

## Author Contributions

**Conceptualization:** Carrie E. Lyons, Jack MacAllister, Greg Millett, Jennifer Sherwood, Stefan D. Baral.

**Data curation:** Nikita Viswasam, Carrie E. Lyons.

**Formal analysis:** Nikita Viswasam.

**Methodology:** Carrie E. Lyons, Amrita Rao.

**Supervision:** Nikita Viswasam, Carrie E. Lyons, Amrita Rao.

**Writing – original draft:** Nikita Viswasam, Carrie E. Lyons.

**Writing – review & editing:** Nikita Viswasam, Carrie E. Lyons, Jack MacAllister, Greg Millett, Jennifer Sherwood, Amrita Rao, Stefan D. Baral.

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
