## [Decision Letter · Decision Letter 0]

7 Nov 2019

PONE-D-19-15539

The Uptake of Population Size Estimation Studies for Key Populations in Guiding HIV Responses in Africa

PLOS ONE

Dear Dr Viswasam,

Thank you for submitting your manuscript to PLOS ONE. After careful consideration, we feel that it has merit but does not fully meet PLOS ONE’s publication criteria as it currently stands. Therefore, we invite you to submit a revised version of the manuscript that addresses the points raised during the review process.

Both reviewers asked you to change the manuscript as it currently is. Please address these for your next iteration. We would appreciate receiving your revised manuscript by Dec 06 2019 11:59PM. To enhance the reproducibility of your results, we recommend that if applicable you deposit your laboratory protocols in protocols.io, where a protocol can be assigned its own identifier (DOI) such that it can be cited independently in the future. For instructions see: http://journals.plos.org/plosone/s/submission-guidelines#loc-laboratory-protocols

We look forward to receiving your revised manuscript.

Kind regards,

Eduard J Beck, PhD, FAFPHM, FFPH, FRCP

Academic Editor

PLOS ONE

**Journal Requirements:**

2. In your Methods section, please provide additional information regarding the methodology used in your systematic review. We strongly recommend that you complete the relevant items of the PRISMA checklist (http://prisma-statement.org/prismastatement/Checklist.aspx) to strengthen the methodology reporting.

In particular, please address the following points:

a) please describe in more detail how PSE were identified (in Phase 1) in the Global.HIV database (in particular, please report the search string used, or similar)

b) please clarify how the 23 different types of documents used in Phase 2 were selected and researched.

Thank you for your attention to these requests.

3. We note that your article has been submitted as a "Collection Review" article type, but is a research article submitted to the Project SOAR Collection. When resubmitting your manuscript, we ask that you update your article type to "Research Article" in the online submission form. Please note that some fields in the submission form, particularly in the "Additional Information" field, will have been reset with this change, so please go through your submission in full to ensure that all information is accurate and complete when resubmitting your manuscript.

**Additional Editor Comments (if provided):**

Reviewer #1:

A sound review that I recommend for publication. It might be worth commenting more on the reasons for the findings such as PSE being often donor driven, and producing underestimations that make the validity of the results questionable and not very useful.

Reviewer #2: It was a pleasure to review the article entitled "The Uptake of Population Size Estimation Studies for Key Populations in Guiding HIV Responses in Africa". While I believe the article is an important contribution to filling an existing gap in literature, I would recommend some clarifications and edits prior to publication.

Major Comments:

1) In Line 200, the authored specified that the focus of the review was on PEPFAR and Global Fund investments. It may be beneficial to bring this fact to the forefront earlier (i.e. perhaps even in the abstract) and even include this in the limitations section.

2) The article could benefit from a greater description of the gray literature sources used. There are many programs that conduct PSE for programming purposes but never publish the estimates in peer-reviewed journals. Were annual reports, programmatic study findings, etc. taken into account?

Minor Comments:

1) Line 174-176: It was unclear which types of articles used in this review (i.e. gray and peer-reviewed).

2) Line 267, 280, 287: Were the PSE used to "change" policy or just to inform it? Were there any instances where targets were not changed because estimates were as expected?
---

## [Author Response · Author response to Decision Letter 0]

16 Dec 2019

Dear Dr. Beck and Reviewers,

Thank you for the review and constructive feedback of the manuscript (PONE-D-19-15539) submitted for consideration in the Project SOAR Collection. Please find below responses to feedback, and an outline of changes made to the manuscript for resubmission.

We have updated the manuscript to meet PLOS ONE’s style requirements following the style templates provided. We have also provided additional information on the methodology of the parent systematic review using the PRISMA Checklist. As part of this addressing point A, we have added more detail on the structure of the Global.HIV data repository as well as the data export and indicator selection process to clarify how PSE were identified in Phase 1 (Lines 178-201). To address point B, we have also added a description of the literature search and selection process of documents in Phase 2 (Lines 250-256).

Responses to Reviewer #1 (R1):

R1 Comment: A sound review that I recommend for publication. It might be worth commenting more on the reasons for the findings such as PSE being often donor driven, and producing underestimations that make the validity of the results questionable and not very useful.

Thank you for the recommendation and suggestion on adding commentary on reasons for findings. We agree that underestimations are a frequent observation that can lead to lack of utility and have noted this in the discussion (Line 461-462). While the motivations of PSE activity funders influencing utilization is an important consideration, evidence in documents conveying the reasoning of PSE funders/implementers was not examined in enough detail to be able to include this as a demonstrable factor influencing research utilization in the discussion section. 

Responses to Reviewer #2 (R2):

R2 Comment: In Line 200, the authored specified that the focus of the review was on PEPFAR and Global Fund investments. It may be beneficial to bring this fact to the forefront earlier (i.e. perhaps even in the abstract) and even include this in the limitations section.

Thank you for this observation, and we have added a mention of this focus on PEPFAR and Global Fund investment in the abstract (Line 73) and as a limitation (Lines 456 – 458).

R2 Comment: The article could benefit from a greater description of the gray literature sources used. There are many programs that conduct PSE for programming purposes but never publish the estimates in peer-reviewed journals. Were annual reports, programmatic study findings, etc. taken into account?

Thank you for this for this important observation, and we have added a description of the gray literature sources used, as well as more detail on the eligibility criteria in the methods. In the Phase I results, we also have added a breakdown on the types of reports in grey literature to better describe their relationship with governmental and nongovernmental organizations (Lines 275-279). We broke down the types of grey literature reports identified, which include those with PSE activities as a component of HIV surveillance studies commissioned in collaboration with governmental organizations, PSE-specific studies, and NGO-led reports.

R2 Comment: Line 174-176: It was unclear which types of articles used in this review (i.e. gray and peer-reviewed).

Thank you for this, and we have added more detail on the eligibility criteria to describe the sources included in the parent systematic review in the methods sections (Lines 174-176) and thus eligible for consideration in the Phase 1 PSE review. These sources include those available on the web from governmental on non-governmental sources (gray literature), as well as including sources published in a peer-reviewed journal or presented as an abstract at a scientific conference.

R2 Comment: Line 267, 280, 287: Were the PSE used to "change" policy or just to inform it? Were there any instances where targets were not changed because estimates were as expected?

Thanks for this important consideration and opportunity to convey how PSE references specifically relate to the ways program planning has been described. In most documents where we reported that PSE was used to change a policy or program, PSE rarely changed policy, but did change programs as described in the actions taken for planning of programs. About a third of documents that referenced PSE data, made no mention of using them to take actions in planning or any language commenting on the results of the estimates (Lines 377-380). But where action has been taken and attributed to the existence of PSE data, the document language specifies a different direction in planning regardless of the results of the estimates. We consider this to be a change in program planning rather than just informing it, and have observed this to be independent of whether or not the results of the estimates were as expected.

---

## [Decision Letter · Decision Letter 1]

22 Jan 2020

The Uptake of Population Size Estimation Studies for Key Populations in Guiding HIV Responses on the African Continent

PONE-D-19-15539R1

Dear Dr. Viswasam,

We are pleased to inform you that your manuscript has been judged scientifically suitable for publication and will be formally accepted for publication once it complies with all outstanding technical requirements.

With kind regards,

Eduard J Beck, PhD, FAFPHM, FFPH, FRCP

Academic Editor

PLOS ONE

Additional Editor Comments (optional):

Reviewers' comments:

Reviewer's Responses to Questions

**Comments to the Author**

1. If the authors have adequately addressed your comments raised in a previous round of review and you feel that this manuscript is now acceptable for publication, you may indicate that here to bypass the “Comments to the Author” section, enter your conflict of interest statement in the “Confidential to Editor” section, and submit your "Accept" recommendation.

Reviewer #1: All comments have been addressed

2. Is the manuscript technically sound, and do the data support the conclusions?

Reviewer #1: Yes

3. Has the statistical analysis been performed appropriately and rigorously? 

Reviewer #1: Yes

4. Have the authors made all data underlying the findings in their manuscript fully available?

Reviewer #1: Yes

5. Is the manuscript presented in an intelligible fashion and written in standard English?

Reviewer #1: Yes

6. Review Comments to the Author

Reviewer #1: The authors have satisfactorily responded to all my observations/suggestions. The article is relevant, concise, clear and informative.

7. PLOS authors have the option to publish the peer review history of their article (what does this mean?). If published, this will include your full peer review and any attached files.

Reviewer #1: Yes: John Waters

---

## [Editor Report · Acceptance letter]

10 Feb 2020

PONE-D-19-15539R1 

The Uptake of Population Size Estimation Studies for Key Populations in Guiding HIV Responses on the African Continent 

Dear Dr. Viswasam:

I am pleased to inform you that your manuscript has been deemed suitable for publication in PLOS ONE. Congratulations! Your manuscript is now with our production department. 

With kind regards,

on behalf of

Dr. Eduard J Beck 

Academic Editor

PLOS ONE